# The Lightning Jump Algorithm for Nowcasting Convective Rainfall in Catalonia

**Carme Farnell** *,† [ID] **and Tomeu Rigo** † [ID]

Meteorological Service of Catalonia, C/Berlin, 38–46, CP 08029 Barcelona, Spain; tomeu.rigo@gencat.cat
* Correspondence: carme.farnell@gencat.cat
† These authors contributed equally to this work.

**Abstract:** Previous studies in Catalonia (NE Iberian Peninsula) showed a direct relationship between the Lightning Jump (LJ) and severe weather, from the study of different events, occurring in the last few years in this region. This research goes a step beyond by studying the relationship between LJ and heavy rainfall, considering different criteria. It selects those episodes exceeding the 40 mm/h threshold, dividing them between those with or without LJ occurrence (3760 and 14,238 cases, respectively). The time and distance criteria (<150 km and <50 min, respectively) allow the detection of rainfall episodes with LJ, to establish an accurate relationship between the jump and the heavy rain occurrence. Then, lightning and radar data are analyzed, considering monthly and hourly distributions. Skill scores for the period 2013–2018 showed good results, especially in summer, with values of POD ≃ 90% and FAR ≃ 10%

**Keywords:** heavy rainfall; lightning jump; weather radar; lightning data

## 1. Introduction

Several authors around the world have studied the behavior of the Lightning Jump (LJ) algorithm to observe the relationship between this and severe weather [1–5]. In the case of Catalonia, the area of study, severe weather is understood as the presence of at least one of these phenomena in a thunderstorm: hail of diameter ≥ 2 cm, wind gusts with speeds ≥ 25 m/s, and tornadoes [4]. On the other hand, the LJ is a sudden increase in the total lightning flash rate [6] and tends to precede severe weather occurrences on the ground [1,6–12]. The sudden increase in lightning activity is believed to be a response to the rapid intensification of the updraft that leads to an increasing number of ice particle collisions and thus greater charge separation and flash rates [2,3,13–15]. However, the research of [16] investigated the correlation between kinematic and microphysical radar parameters and lightning flash rates. They found that maximum updraft speed had the lowest relationship with flash rate, while graupel mass fitted better with flash rate. These results showed that the microphysics of thunderstorms have still some limitations. Therefore, more studies are necessary to improve this point.

Regarding the area of study (Catalonia, NE Iberian Peninsula), some studies also proved the relationship between LJ and severe weather. In this sense, the work in [17] was the precursor analysis of a list of works that analyzed particular cases. One of the most outstanding studies was the one of [18], which observed an increase in the number of flashes in a tornadic event. However, it was [4] who systematically studied the behavior of the LJ algorithm as a tool for forecasting severe thunderstorms, analyzing a set of more than 40 cases that occurred during the period 2006–2013. The results obtained can be summarized as the Percent Correct (PC, which is defined as the verification measure of the performance, consisting of the rate of the total number of correct forecasts and the total number of predictions) > 73% and a false alarm ratio (FAR) < 10%. Continuing the previous research, the work in [5] analyzed some lightning and radar parameters associated with those LJ-producing

thunderstorms, to search for some key elements that could help to forecast the type of severe weather. In this case, the results were not particularly accurate, showing some trends, but not conclusive at all. In any case, and based on the work of [4], the Meteorological Service of Catalonia (SMC) put in operation and currently uses this algorithm for surveillance tasks, allowing warning the population via the civil protection authority in case severe weather could affect a strategic location.

Although the LJ was initially related to severe weather, in the last few years, some works have tried to associate it with convective rainfall (this, with the peak of maximum intensity or one hour cumulated rainfall, depends on the analysis and the data available). For instance, some studies have shown a positive relationship between the increase of the lightning activity (searching or not for jumps) and the intensification of precipitation, with a lead time (LT), which is the time gap between both occurrences, that moves from a few minutes to one hour [19–25]. The work in [26] showed that the main connection between lightning and rainfall was due to rebounding collisions between ice-phase particles in the presence of supercooled liquid water, which is related to non-inductive charging and cold-cloud precipitation, depending on the type of precipitation [27,28]. One part of this type of precipitation is the collision-coalescence formed by the water vapor condensate on aerosol particles, which form small drops that compose cloud water. This condensate is then collected by large drops that eventually reach the ground [29]. In the region, some papers [30,31] studied the relationship between lightning and precipitation in summer thunderstorms, but without considering the LJ occurrence. Furthermore, from the authors and other SMC technicians' experience, in the last few years of operational application of the LJ tool, it has been observed that most of the severe weather events had also included brief and large rainfall intensities.

In this manuscript, the behavior of the LJ in front of heavy rainfall in Catalonia is methodically studied, in order to analyze the performance of the algorithm in this type of event, to use it to improve warning techniques for the population. To achieve this objective, some parameters associated with the precipitating clouds are analyzed. The paper is divided into the following parts. Firstly, we delimit the area of study. Then, we define the data used and the methodology. The next step is the presentation of the results, which are analyzed in the discussion and summarized in the conclusions.

## 2. Data and Methodology

### 2.1. Area of Study

Catalonia (NE Iberian Peninsula) is a region of 32,000 km$^2$ with a complex territory (Figure 1). Different ranges run across the area: firstly, the Pyrenees and the Pre-Pyrenees are quasi-parallel, having an E-W orientation, and maximum heights over 3000 m and 2500 m, respectively. Besides, the pre-littoral and littoral ranges move also quasi-parallel from SW to NE, with lower altitudes (over 1500 m and 1000 m, sequentially). Furthermore, it is important to remark, as happens in many other Mediterranean regions, that in most of the cases, these ranges present steep slopes, which could help the triggering of convection in some concrete environments. Other important features are several more or less deep valleys and plains, located between the orographic ranges. Besides, the coast runs approximately from SW to NE. In this sense, the Mediterranean Sea plays a significant role in the climate of the region, regulating the temperatures and providing high amounts of moisture.

Different air masses can affect the study area, mainly from the Atlantic (maritime, cold, and polar), from Central-North Europe (continental, cold, and polar), or from Northern Africa (continental, warm, and tropical). This and the topography contribute to the occurrence of many severe weather events and heavy rain episodes in Catalonia. Some of these events can produce extended floods or local flash-floods [32–35]. Most of the flash-floods are due to deep convective systems [36,37]) similar to those that produce severe weather. The analysis in [38] and other authors have still reported that heavy rainfall and severe phenomena appeared simultaneously in some episodes.

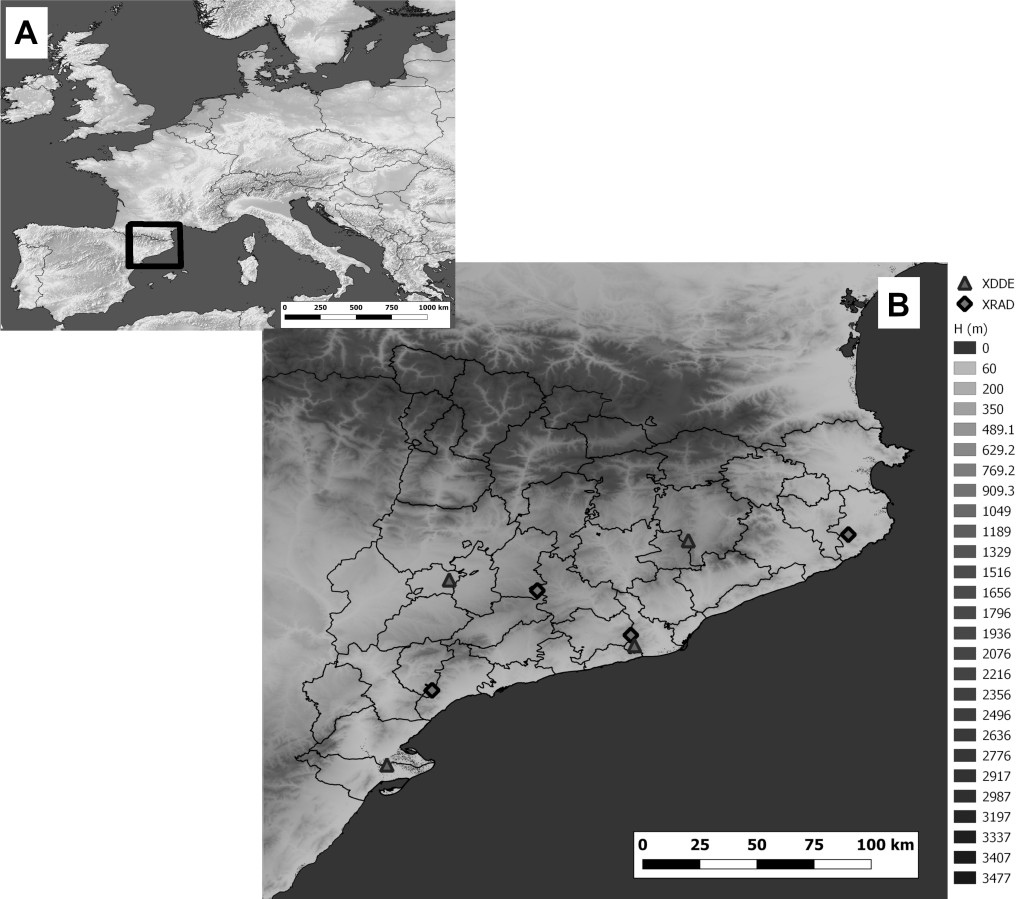

**Figure 1.** (**A**) General view of Western Europe. The black rectangle marks the study zone; (**B**) zoom view of Catalonia, the area of analysis, including the locations of the weather radars (diamonds) and lightning detectors (triangles).

*2.2. Data*

This section introduces the different types of networks, the data produced, and the products generated, used in some of the steps of the analysis of the relationship between heavy rainfall and lightning jump occurrence.

2.2.1. Automatic Weather Station Network

The Servei Meteorològic de Catalunya (SMC or Meteorological Service of Catalonia) manages a network of Automatic Weather Stations (AWS) called XEMA. It is composed of 180 elements, which covers a large part of the territory and minimizes the impact of the inhomogeneities in the local weather. However, it presents a disadvantage: these elements come from different origins: from the Agricultural Department to the Catalan Agency of Water management, plus others that have been installed directly by the SMC. Because of this, there are some differences in the type of sensors and data collected. However, one of the current objectives of the SMC is to unify all the stations in the future. The main differences are the height of the wind sensors (2, 6, and 10 m) or how the rainfall is measured and recorded. In any case, and thanks to the Quality Control Unit, the final data show a good homogeneity [39].

Most of the AWS is capable of recording 1 min rainfall, which easily allows generating a 30 min moving window accumulation. However, a few stations (20 of the 180 elements) only record precipitation half-hourly, at fixed minutes (this is, at 00 and 30 min -00:00, 00:30, 01:00, 01:30-). As we will show later, the AWS data were combined with radar data to generate more realistic hourly rainfall

fields. Because the analysis is focused on the occurrence of heavy rainfall in a short time, all those cases that exceeded the 40 mm/1 h threshold were detected.

### 2.2.2. Lightning Location System

The SMC had operates since 2004 a Lightning Location System (LLS) (XDDE from the acronym in Catalan). XDDE combines the detection of cloud-to-ground (CG) and intra-cloud (IC) flashes, using Low Frequency (LF) and Very High Frequency (VHF) sensors, respectively. Then, each sensor detects the two flash types separately. The network is composed of four Vaisala LS-8000 detectors (Figure 1), strategically located to obtain good coverage over all the territory of Catalonia and the surrounding areas.

The system processes the IC and CG flashes separately. On the one hand, it locates IC flashes using the interferometry technique [40]. The baseline of the sensors (135–150 km) does not allow finding flashes' altitudes [41]. The LLS locates a maximum of 100 s-1events in steps with a resolution of 100-μs. Two successive VHF events correspond to the same IC flash if there is less than 10 km and 0.5 s between them.

On the other hand, Low Frequency (LF) sensors detect CG strokes. The system locates the strokes using a combination of the time-of-arrival and magnetic direction-finding techniques [42]. CG flashes group those CG strokes based on time and distance criteria: with a maximum interval between strokes of 0.5 s, a maximum flash duration of 1 s, and a spatial radius limited to 10 km [42]. Total Lightning (TL) is the sum of IC and CG flashes.

Successive campaigns using electromagnetic field measurements and video recordings of natural lightning allow evaluating XDDE performance experimentally [43–45]. These analyses established a Detection Efficiency (DE) of around 80–85% for CG flashes and around 70–75% for IC flashes. The estimated median location accuracy is ∼1 km for the CG strokes and a bit more for the IC events.

In the present study, lightning data were used in two ways: the first one was the study of the IC and CG yearly distributions and the comparison of the different behaviors; secondly, the lightning data were the source for the triggering of warnings of the LJ algorithm (more information in [4]). It is worthy noting that the current analysis presents an important difference with the previous LJ studies [4,5], in which the algorithm runs only over TL flashes (that is, one point per lightning event). In the last few years, a new complementary system has added to the operational procedure. This new module considers all the events associated with an IC flash and all the strokes included in a CG flash (that is, several points per lightning flash). This increase in the number of registers favors a better thunderstorm detection. Besides, it also helps to maintain the criteria of space and time in the convective cells' tracking. However, experience has shown that this new procedure detects many thunderstorms that do not produce severe weather phenomena. On the other hand, most of those thunderstorms lead to heavy rainfall situations. This fact encouraged starting the current analysis. The results in the present study show the combination of both systems: only one point per flash and multiple points per flash. In this sense, the analysis considers all the warnings of both types (the percentage of warning with multiple points is 74.5%, corresponding to a total of 3307 cases). Finally, the work in [4] defined the LJ criterion used in this work. A warning happens in a thunderstorm when the last minute total lightning rate is higher than a threshold. This value corresponds to double the standard deviation for the previous 14 min.

### 2.2.3. XRAD: Radar Network

The Radar Network (XRAD) of the SMC is composed of 4 C-band single-pol Doppler weather radars (Figure 1). One of the main purposes of the XRAD is to provide high-resolution quantitative precipitation estimations (QPE), among other meteorological and hydrological products [46]. The main advantage of the composition is that the negative effects over the radar imagery (e.g., signal attenuation or signal blockage) are partially mitigated [47]. Because of the good spatial distribution, ninety-six percent of the whole Catalan territory is covered by at least one volumetric raw, while 58% is covered

by the scan of two or more radars (SMC internal report). The other main purpose of the XRAD is to help in the surveillance and forecasting tasks in adverse weather events (from snowfalls at low heights to severe thunderstorms). One of the most challenging aspects related to radar information is the difficulty to identify patterns related to severe weather, because of the complexity of the shapes in the radar imagery (see, for instance, [48,49]). Therefore, the combination of radar imagery with lightning data results in being providential in order to analyze the nowcasting of severe weather events, but also heavy rainfall episodes [21].

The use of weather radar data has two purposes in this work. One is the use of the hourly rainfall estimation, generated by combining the radar data (previously corrected) with rain gauge values as [50] described. This combination of both sources allows maintaining the ground registers and, on the other hand, the radar shape of the precipitation field. This product has made the task of the rainfall maxima identification easier, which the rain gauge has not been able to detect. On the other hand, different radar products have allowed finding the characteristics of the thunderstorms during the period between the LJ warning and the rainfall register occurrence. These data are provided by the life-cycle tracked using the operational algorithm of the SMC (for more information, see [31,51]).

*2.3. Methodology*

To speed up the selection of episodes to study, a general criterion was applied to discriminate days without precipitation or lightning activity. For this reason, two thresholds were considered in the study of the daily information of the period 2013–2018:

(i)　　Accumulated precipitation/day　3 mm
(ii)　　Number of CG flashes/day　25

Those thresholds were selected in concordance with different criteria. All of them departing from the experience of the authors in operational tasks, in which it was observed that over those values, it an event without real precipitation or thunderstorm activity during the period 2012–2019 never occurred. Besides, in some days with anomalies in the radar or the rain gauge networks, the quantitatively estimated precipitation product could show non-null values, which were not easy to remove automatically. A similar pattern could appear in the daily lightning map, associated with the bad functioning of a sensor. In any case, those events with real precipitation or lightning activity with values under those thresholds are scarce in number. The previous criteria allowed the following classification (Table 1).

**Table 1.** Different categories in which all the days of the period of study are labeled, considering the precipitation and the lightning activity registered.

| Day | | CG Flashes > 25 | |
|---|---|---|---|
| | | YES | NO |
| QPE > 3 mm | YES | rainy, convective | rainy, but not convective |
| | NO | dry, convective | dry, non-convective |

Since here, the analysis of the convective rainy days consisted of the following steps:

1.　　Hourly precipitation: The process starts selecting those pixels with hourly precipitation $> 40$ mm in the QPE fields combining radar and rain gauges. The size of the pixels is $1 \times 1$ km$^2$. Those pixels with QPE under this threshold are labeled as null, maintaining only the regions exceeding the cited value. From here, we define as an event (or *QPE* cell) each one of those pixel regions with

rainfall exceeding 40 mm/h in the estimation product generated by combining radar and AWS data. The process of event selection is presented in the top panels and the left one at the bottom of Figure 2. All the consecutive pixels in a rainfall cell were grouped, defining the area and centroid. The equation of the mass centroid allowed the calculation of the centroid position ($x_c$, $y_c$) (see Equations (1) and (2), where $QPE_i$ is the estimated precipitation at the $i^{\text{th}}$ pixel and $x_i$ and $y_i$ are the coordinates of the same pixel). Once the area of the rain cell is estimated, it is possible to determine the time when the maximum rainfall has occurred. The analysis of the reflectivity field allows finding the time, searching for the best correlation in space between the convective cell and the QPE region centroid. The time gap between consecutive imagery was 6 min. The central and right bottom panels of Figure 2 show an example of the procedure. In the presented case, the time of the maximum intensity for the QPE Areas C1 and C2 was 10:00 and 10:36, respectively.

$$x_c = \frac{\sum_{i=1}^{N} QPE_i * x_i}{\sum_{i=1}^{N} QPE_i} \tag{1}$$

$$y_c = \frac{\sum_{i=1}^{N} QPE_i * y_i}{\sum_{i=1}^{N} QPE_i} \tag{2}$$

2. Relationship between hourly precipitation with LJ or without LJ (woLJ): For each event considered in the previous step, the *QPE* cells were selected and searched for the presence or not of one or more closed LJ. To link both phenomena (QPE cell and LJ), they must occur in a spatial distance < 50 km and a temporal range < 150 min. Those thresholds from the authors' experience in 4 years of operational application of the algorithm were selected. Although not usual, it is possible to detect some LJ warnings in thunderstorms that have been produced 2 h or more before the occurrence of the event at the ground. Besides, the LJ notification is registered before (some minutes) the rainfall rate peak, in some particular cases. Continuing with the same situation of Figures 2 and 3 shows the QPE cells C1, C2, and C3 and the LJ L1, L2, and L3 (in all events, the time on the right or at the bottom indicates the time of the maximum rainfall or the occurrence of the LJ, respectively). In the example of the figure, L2 and C3 are related. C3 initially was associated with two LJ, L2 and L3. Both warnings agreed with the restriction of distance, but only the first one verified the condition of time. On the other hand, L1 and L3 and C1 and C2 were independent phenomena because they were not supported by the time and space criteria. It is necessary to clarify that one LJ can be associated with more than one cell if both verify the previous rules. Once all the QPE cells of the full period have been labeled as LJ or woLJ, it is possible to start the other steps of the analysis. We classified the cells into three categories. First, $ev_{LJ}$ were those cells associated with at least one LJ (in Figure 3, cell C3). $ev_{prLJ}$ refers to those QPE cells that took place during events with LJ, but they could not be linked with any warning because some of the spatial and time constraints were not verified (cells C1 and C2 in the same figure). Finally, $ev_{noLJ}$ occurred without any LJ on the same day. The first part of the analysis considered this classification (Sections 3.1 and 3.2). On the other hand, a unique class ($ev_{noLJ}$) included the $ev_{prLJ}$ and $ev_{noLJ}$ categories in the last part of the study (Sections 3.3 and 3.4). This change was because it only mattered if an LJ-cell relationship in the forecasting tasks existed or not.

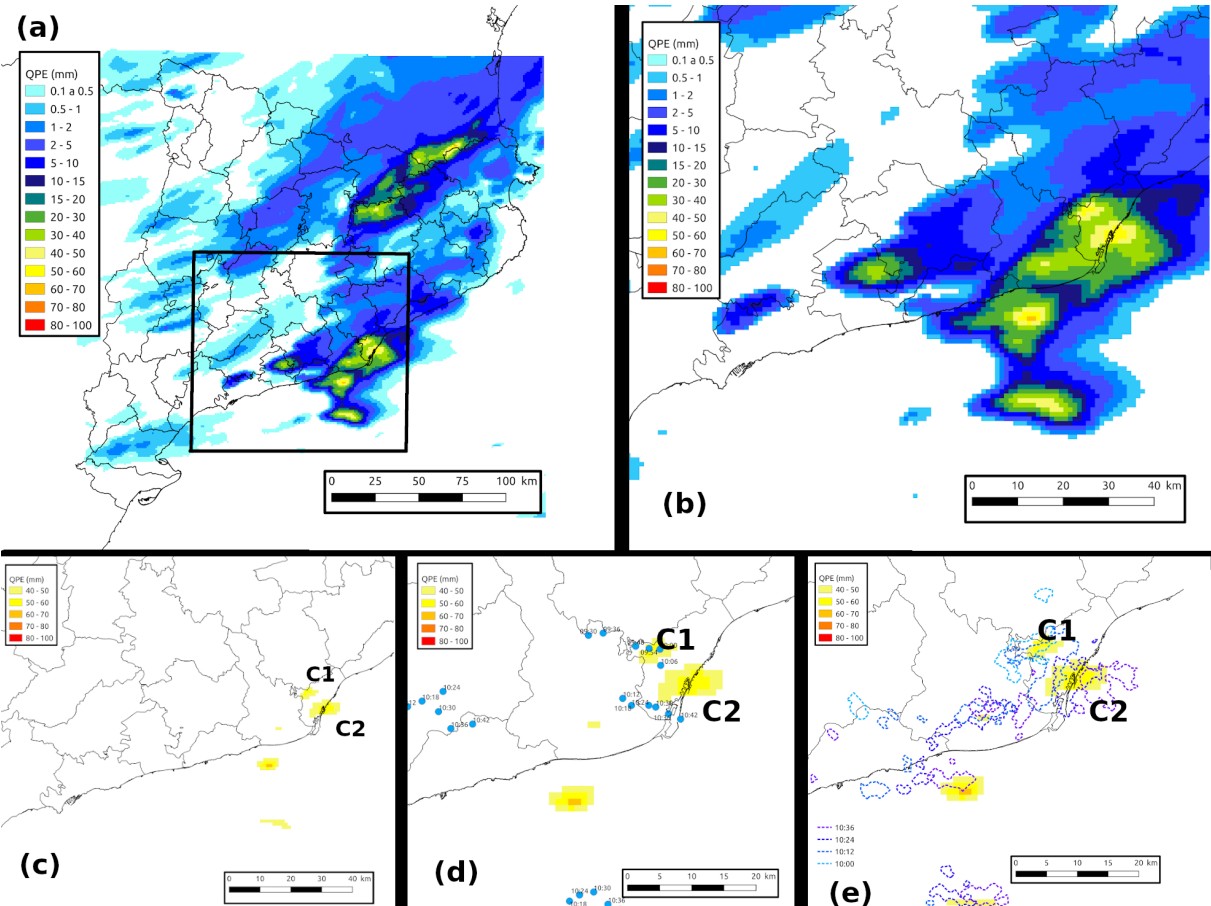

**Figure 2.** Process of the selection of the rainfall structures and assignation of the time, for the example of 2019/07/27 10:00 UTC: (**a**) Total hourly QPE field considering the radar and AWSs' data, for the whole Catalan region (the black square shows the zoom to the area of analysis in the example). (**b**) Total hourly QPE field zoomed. (**c**) Converting pixels with QPE < 40 mm to Null. QPE cells C1 and C2 are labeled for the analysis shown in the following panels. (**d**) Comparison of the QPE areas of analysis (yellow ones, C1 and C2) with the centroids of the 6 min convective cells (blue dots) (the labels indicate the time of occurrence) in a zoomed region of the Central Coast of Catalonia. (**e**) The same as (**d**), but with the 6 min reflectivity regions exceeding 45dBZ (dashed line), at different times (colors indicate the hour and minute).

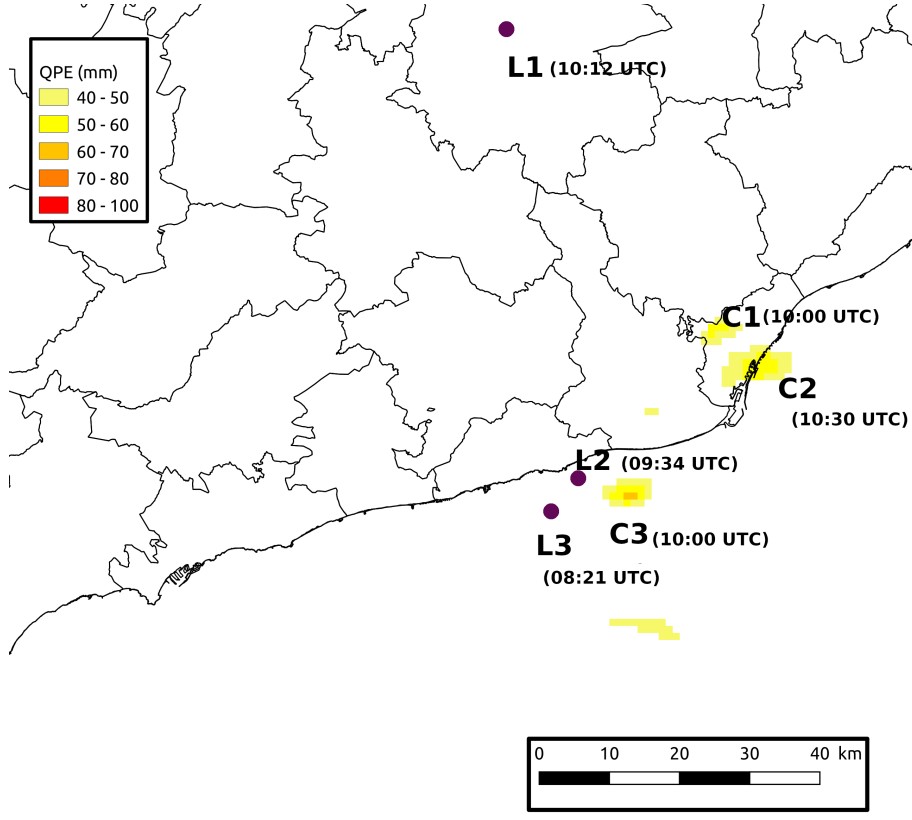

**Figure 3.** Criteria applied to connect the LJ (L1, L2, and L3) to the heavy rainfall cells (C1, C2, and C3). See the text for more information.

3.  Characteristics of lightning activity: All the lightning recorded at a distance less than 50 km with respect to the precipitation's centroid and 2 h before and after the maximum rainfall value were selected (see Figure 4). The number of IC, +CG, and –CG flashes was calculated for each QPE system, to observe the annual and monthly behavior and to compare the number of flashes between systems with and without LJ.

4.  Characteristics of weather radar detected convective cells: A similar criterion as used in Point 3 (characteristics of lightning activity) was applied to the radar characteristics (Figure 5). The convective cells selected in the study needed to be detected closer than 25 km with respect to the QPE centroid's point and, also, a maximum of 2 h or later before the maximum rainfall instant. Through this selection, different radar parameters were studied such as TOP (maximum height of the thunderstorm), VIL (Vertical Integrated Liquid), and the height centroid (that is, the altitude where the mass center of the thunderstorm was located, considering the reflectivity instead of the mass), among others.

5.  Monthly and spatial distribution of the precipitation: The last step of the present analysis resulted in the monthly and spatial distribution of the QPE cells, to make a better characterization of them. In this case, both groups (with and without LJ) were analyzed. Furthermore, we evaluated the capability of the LJ for forecasting heavy rain events for the same time distributions, by means of different skill scores.

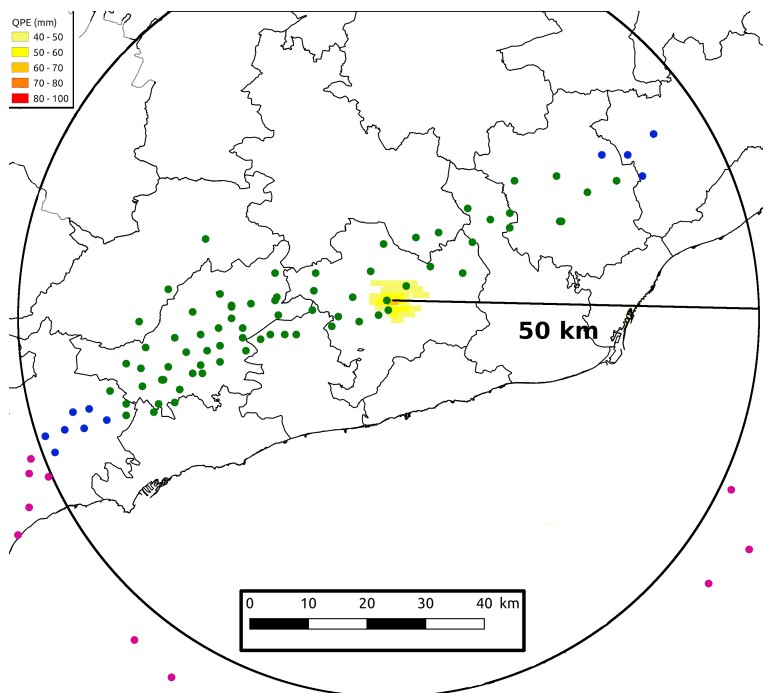

**Figure 4.** Criteria applied to connect CG flashes with the QPE centroid: green dots are those CG flashes that verify both criteria (time and space); blue dots are not valid because of the time with respect to the maxima rainfall moment; and purple dots are those not valid because of the criteria of space (or both). The black circle indicates the 50 km distance to the centroid of the QPE system.

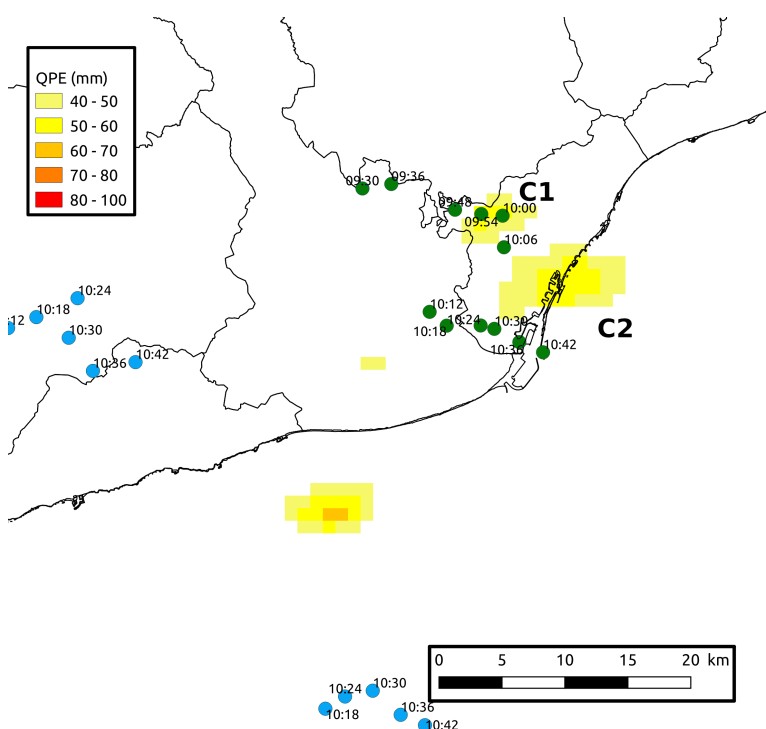

**Figure 5.** Criteria applied to connect radar convective cells with the QPE centroid: Green dots are those cells associated both systems C1 and C2 (because both were caused by the same rain system, it was impossible to discriminate if one or other thunderstorm was involved in both QPE cells). On the other hand, blue cells are rejected as contributors to C1 or C2.

## 3. Results

The analysis of the period 2013–2018 produced the detection of a total of 17,998 events (areas with estimated rainfall exceeding 40 mm). Of these, 3760 had associated at least one LJ ($ev_{LJ}$); 10,408 QPE cells took place during events with LJ, but they could not be linked with any warning because some of the spatial and time constraints were not verified ($ev_{prLJ}$); and, finally, 3830 occurred without any LJ during the same day ($ev_{noLJ}$). In this section, some results of the study of the set of QPE cells, mainly those with LJ associated, are presented.

### 3.1. Monthly, Hourly, and Spatial Distribution of the Events

According to the classification of the events (with LJ, $ev_{LJ}$, without linked LJ, $ev_{prLJ}$, and without LJ, $ev_{noLJ}$), the first behavior to be considered was the distribution by months and hours of the three sets of QPE cells. In the case of the two first categories, the monthly distribution was quite similar (see Figure 6), with a scarce probability of occurrence between January to April and October to December (10.9% for $ev_{prLJ}$ and only 8.0% for $ev_{LJ}$) and a maximum comprised of June to August (65.2% for $ev_{prLJ}$ and 66.7% for $ev_{LJ}$) coinciding with warm months. During this period, the high temperatures were favorable to developing clouds due to convection conditions. In the case of $ev_{noLJ}$, the distribution varied notably, with 60.1% of the cases spread between September and October, while for the rest of the months, only May (5.8%), June (6.6%), and November (8.0%) exceeded 5%.

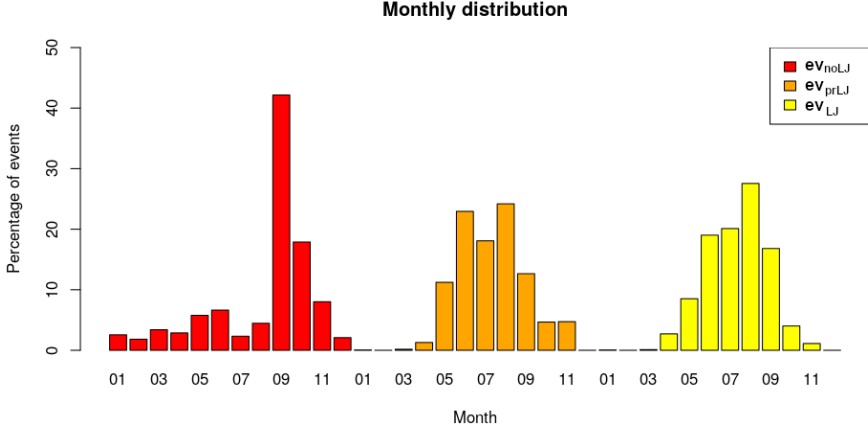

**Figure 6.** Monthly distribution for each one of the event categories (with LJ, $ev_{LJ}$, with no linked LJ, $ev_{prLJ}$, and without LJ, $ev_{noLJ}$).

Moving to the hourly distribution (Figure 7), eighty-one-point-five percent of the cases of the $ev_{LJ}$ events occurred between 11 and 20 UTC, with a clear maxima between 13 and 16 UTC (more than 10% during each hour). In the case of the QPE cells $ev_{prLJ}$, the maxima were slightly displaced to an earlier part of the day, with 64.6% between 11 and 17 UTC and between 12 and 14 UTC with more than 10% of cases in each hour. As happened with the monthly distribution, the time occurrence of $ev_{noLJ}$ events was again more evenly dispersed, without any hour exceeding 10%: from 12 to 17 UTC, the values moved between 7.3% and 9.8%, the rest giving out 49.5%.

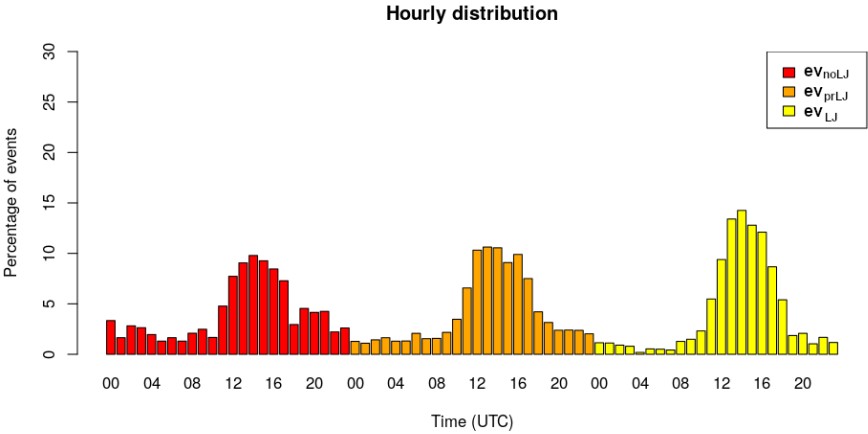

**Figure 7.** Same as Figure 6, for the hourly distribution.

Regarding the spatial distribution, the first result obtained from Figure 8 was the high variability depending on the season of the year and the type of event. The figure shows the more representative months for each type of event ($ev_{NOLJ}$, top row, $ev_{PRLJ}$, middle row, and $ev_{LJ}$, bottom row). The reader must note that in the case of $ev_{NOLJ}$, only two months were representative (September and October; the rest did not have enough cases to show a clear pattern), while in the other two categories, only the more illustrating ones are shown. The first result was the influence of the Mediterranean Sea in the distribution. In the case of $ev_{NOLJ}$, there was a high number of these event types occurring in the coastal and over the Sea regions (top panels of Figure 8). This was because during this period (in September and October), the temperature of the Mediterranean Sea's water remains high and contributes to the convection. It is worth noting that the spatial distribution refers to the QPE cells. Besides, the lightning and LJ warning monthly distributions presented similar patterns to the bottom row [52,53]. The previous charts (Figures 6–8) help to understand the usefulness of the tool. The LJ algorithm resulted in being very helpful for surveillance tasks during summer, afternoon, and inland events. On the contrary, as the lightning activity decreased notably, this is for coastal events during autumn, the precision of the tool decreased. For this type of case, other procedures are necessary for forecasting the heavy precipitation. This electrical behavior coincided with that observed in other regions [54,55]. The contrast of temperatures between the sea and the air favored the severity of the thunderstorms, and furthermore, the content of water was an important factor to be considered. The behavior of $ev_{PRLJ}$ (see the middle panel of Figure 8) could be placed between the two other categories, with two areas (the mountainous regions of the south close to the sea and the sea area just in front of the central coast) that presented high values of occurrence. However, these values were lower with respect to the maxima obtained for the Pyrenees area, especially if we compare with the results obtained for $ev_{NOLJ}$. On the contrary, most of the heavy rainfall $ev_{LJ}$ was produced in the mountain areas of the central-north of the region (bottom panel of Figure 8) due to the essential role the terrain plays in triggering the development of thunderstorms. In this case, the sea seemed not to play any role in the occurrence of the events.

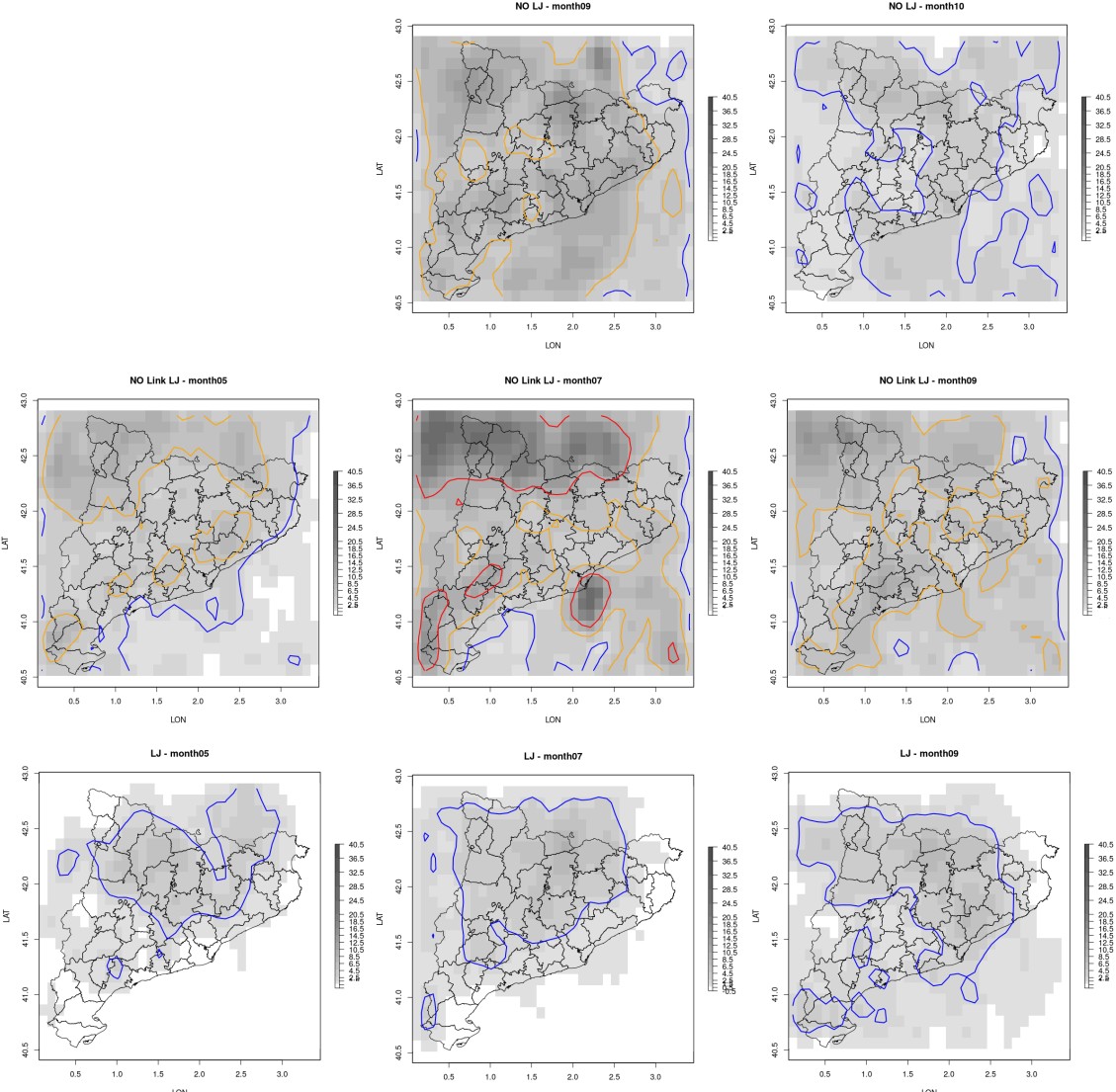

**Figure 8.** Monthly spatial distribution of the most representative cases of QPE cells without LJ (**top**), with non-linked LJ (**center**), and with LJ (**bottom**). Color lines indicate the limits of the areas with values over 2 (blue), 5 (orange), and 10 (red) events per month.

*3.2. Monthly Distribution of Flashes Associated with QPE Cells*

Figure 9 summarizes the lightning behavior associated with the three categories of QPE rain cells. With respect to the intra-cloud (IC) flashes, there were two main points. First, the median (white dots) of the monthly populations in the events associated with LJ was more constant (and concentrated in the period April to November) than the other two categories (more probable during all the year, especially in the case of $ev_{noLJ}$ events). The second point was that the $ev_{LJ}$ median (equivalent to about 35,800 flashes per event) was notably higher than for the other two types (2400 flashes/event for $ev_{noLJ}$ and 3700 for $ev_{prLJ}$).

Moving to the positive and negative CG, the $ev_{LJ}$ QPE cells presented median values over those observed for the other two classes, $ev_{noLJ}$ and $ev_{prLJ}$ events. In the first class, registers were 60 positive and 300 negative flashes/episode. With respect to $ev_{noLJ}$, the values were 10 positive and 35 negative flashes/episode. To conclude, the results for $ev_{prLJ}$ were 15 positive and 50 negative flashes/episode. The previous investigations showed the highest electrical activity in rainfall-precipitating structures associated with LJ occurrence. Besides the variability of the median values along the year, this was

also higher in the case of no occurrence of LJ. Finally, a general aspect for all types of flashes was the absence of cases with low values for the category of $ev_{LJ}$, while in the other classes of events, it was possible to register a rainfall event exceeding 40 mm/h with low or null lightning activity. An LJ warning usually occurred with more than 25 flashes per thunderstorm, according to the study of several storms during the operational period. As a consequence, a cell ($ev_{noLJ}$ or $ev_{prLJ}$) had a reduced electrical activity when it produced less than 10 flashes. This definition was not contradictory with the selected criteria presented in Table 1: the table shows daily values, while a cell refers to periods of a few hours. Then, more flashes can occur in other parts of the day, exceeding the daily threshold.

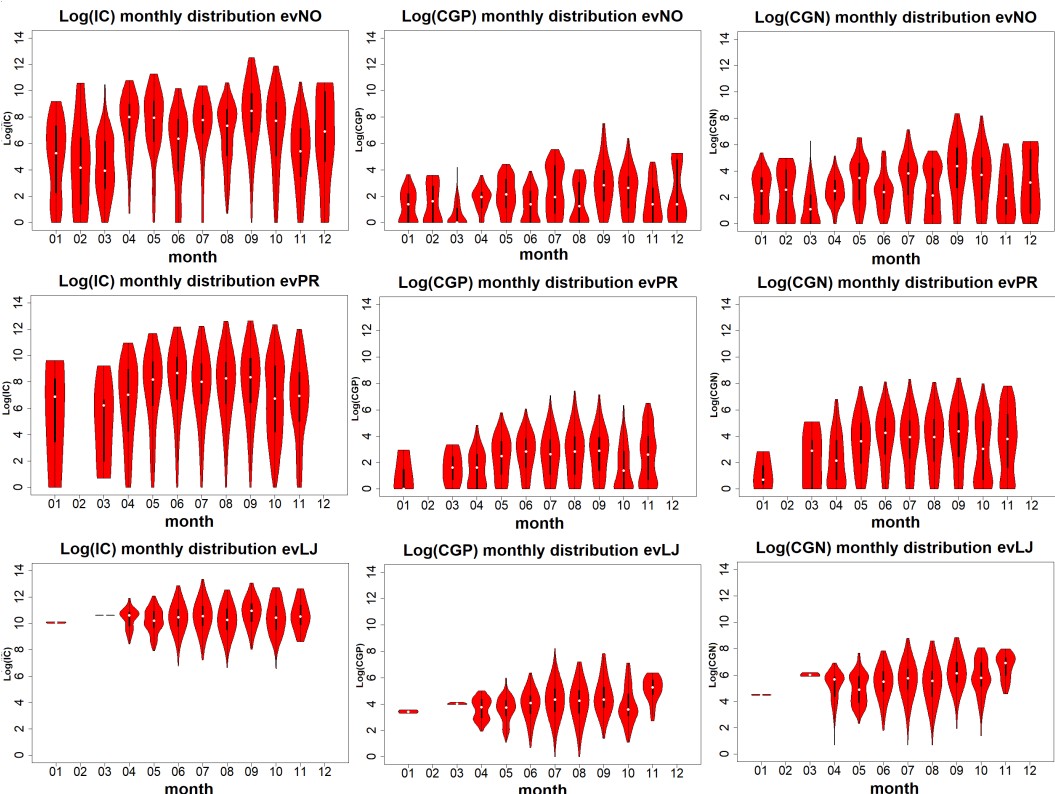

**Figure 9.** Violin plot of the monthly distribution in logarithmic scale of intra-cloud (**left**), positive cloud-to-ground (**center**), and negative cloud-to-ground (**right**) flashes associated the three type of events: without LJ (**top**), with non-linked LJ (**center**), and with LJ (**bottom**).

### 3.3. Radar Parameters of Thunderstorms Associated with Events

At this point, we merged $ev_{noLJ}$ and $ev_{prLJ}$ in the $ev_{noLJ}$ category, to produce discriminators between producing and non-producing LJ warnings. Then, the next analysis only refers to $ev_{noLJ}$ and $ev_{LJ}$, and does not care if the electrical activity in non-producing LJ was elevated, moderate, or limited. Although the number of cases of the first category was notably higher (14,000 cells in front of 3800), the number of radar thunderstorms detected with the operational nowcasting algorithm of the SMC [51] was lower (24,000 for $ev_{noLJ}$ in front of 50,000 for $ev_{LJ}$). The previous values indicated that the median of thunderstorms per $ev_{noLJ}$ cell was two. On the other hand, this rate increased to 13 in the case of $ev_{LJ}$.

In this subsection, some characteristic radar data of $ev_{LJ}$ or $ev_{NOLJ}$ are studied and compared. The work in [56] gave a good hint about most of the radar parameters considered in the present study. It is important to keep in mind that this analysis was done contrasting the results between months, but this statistic could be slightly affected by the density of the sample in each. Firstly, the height of the thunderstorm centroid was analyzed. The median of the majority of $ev_{LJ}$ located its centroid around 2 and 3 km from March to October. Considering the maximums, during the warmest months, from June to September, some centroids arrived at 6 km, while in the other months, the centroids were less than 5 km. In any case, the median values for the whole period were located at a 3 km height. This was caused by convective conditions produced by warm air, which allowed a more significant vertical development (top left image of Figure 10). The same figure shows that the centroids of $ev_{NOLJ}$ were found at slightly lower levels: the median was around 2 km and their maximums were until 5.5 km in July and August. The big difference with the previous group was the observation of the centroid during winter months located between 1 or 2 km (in the case of $ev_{LJ}$, this parameter was null during this season).

The TOP12 and TOP45 (the TOP product considering reflectivity thresholds of 12 and 45 dBZ) of the thunderstorms presented a similar behavior as the height of the centroid, arriving in the case of $ev_{LJ}$ at 15 and 11 km (respectively) during this warmest period (between June and September), while the maximum and median reflectivity obtained 65 dBZ and 44 dBZ (Figure 10, bottom and middle panels, respectively). This was because the majority of these thunderstorms carried solid precipitation inside of them, as was explained by the large values of VIL in some cases (over 20 mm, top-right panel of Figure 10). In fact, and considering the theory of non-inductive electrification [27,28], all these reasons would explain why there were many flashes during the warmest months; in other words, the big vertical development of the thunderstorms and the presence of ice particles in the clouds favored the presence of lightning. In the same way, it is important to remark that during the winter months, we did not find thunderstorms with LJ.

As an example of the monthly evolution, we went in depth with the VIL (Vertical Integrated Liquid), which followed the same shape as the rest of the parameters: it was higher during summer months, especially in the case of $ev_{LJ}$, reaching in some thunderstorms values of 60 mm (related to the presence of solid ice particles). On the contrary, in $ev_{NOLJ}$, the contents of VIL were similar during the entire year, moving between 5 and 10 mm. For all the parameters, the values were limited by the thresholds indicated in Figure 10.

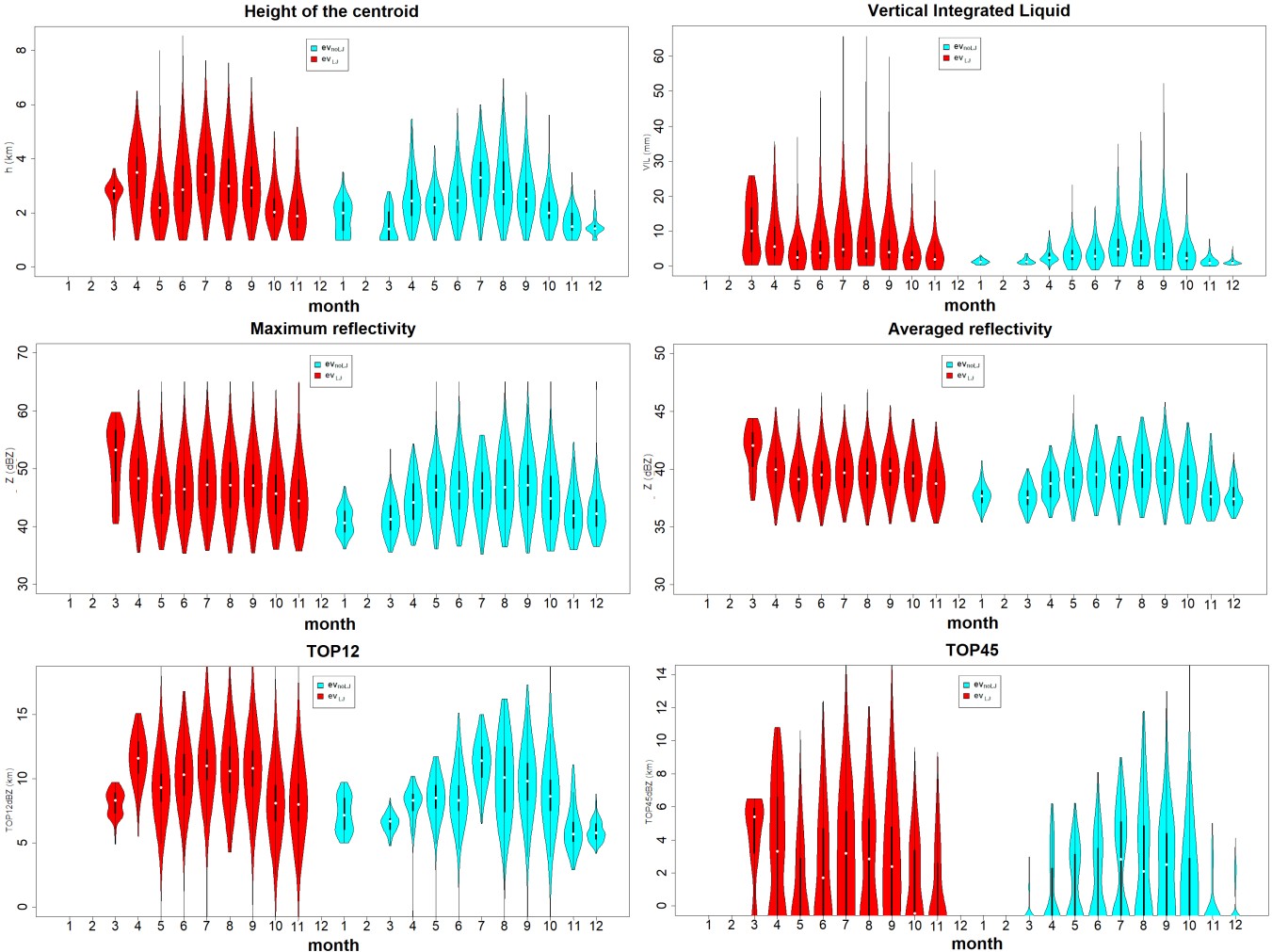

**Figure 10.** Violin plot of the monthly distribution of different radar parameters' comparison for LJ (red) and non-LJ (light-blue) events. From top to bottom and from left to right: centroid height, VIL, maximum and averaged reflectivity, and TOP12 and TOP45.

### 3.4. Validation of the LJ as a Tool to Forecast Heavy Rainfall.

The dataset selected for the skill scores analysis considered the full list of events. Each event corresponded to an observation. The forecasting was the occurrence of LJ warnings. The constraints presented in the second point of Section 2.3 delimited the time and space coincidence between observation and forecast. These constraints seem very lax at first look. However, they have their significance at the time of considering operational surveillance purposes in two ways. The first one is the question related to the lead time in thunderstorms producing LJ. Most of them had a long duration (more than one hour; it is possible to reach 4 h), and the warnings generally occurred at the beginning of the life cycle, during the development stage. On the contrary, both severe weather phenomena and heavy rainfall usually occurred just in the transition between mature and dissipating stages. The second operational point was that the knowledge of the probable occurrence of adverse weather in a thunderstorm with several minutes in advance implied a significant milestone for the surveillance tasks. Besides, the operational experience introduced a new condition: in some specific cases, there were many repetitions of cells in a continuous period. The best example of this situation occurred in heavy rainfall situations associated with convective trains [57] moving from sea to coast ([58] presented some cases). In these episodes, the thunderstorms affected the same area recursively, producing several heavy rain cells (as we defined in the Methodology Section). Because of this, were introduced a new constraint that considered as a unique case all those heavy rain cells grouped in time and space (that is, associated with the same radar structure and occurring in less than 1 h).

The POD (Probability Of Detection) and FAR (False Alarm) skill scores [59] were used to validate the LJ as a tool for forecasting heavy rainfall events (>40 mm in one hour). The POD parameter moved between zero (bad quality of the forecast) and zero (perfect forecast, or all the warnings correspond with an occurrence). In this study, higher values of POD corresponded to the period from April to October (>0.8), with the highest values from June to August (>0.95), coinciding with the warmest months. In the case of FAR, the values went also from zero to one, but with the opposite behavior (this meant that values close to zero implied good results). This parameter showed in general low values (good quality, that is few cases of warning implied a false alarm), overall during the warmest month. Then, these type of thunderstorms were formed by convective conditions, which produced, sometimes, severe weather. In this sense, the analysis in [4] showed, also, how the LJ was a good tool to forecast severe weather (Figure 11).

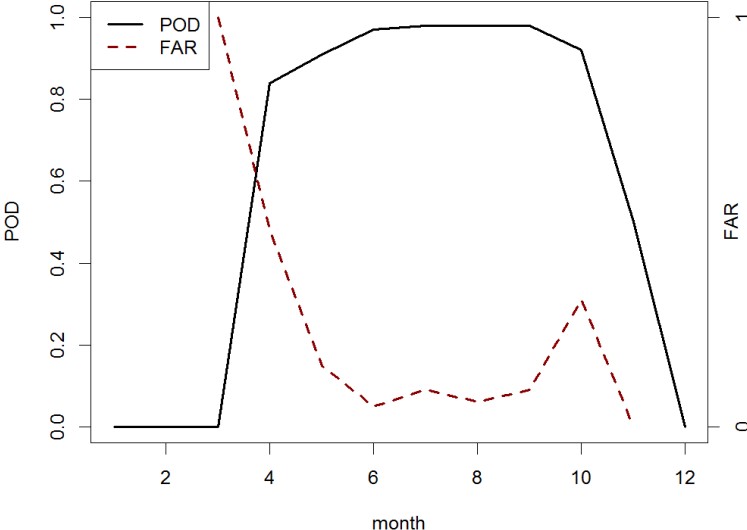

**Figure 11.** Monthly evolution of POD and FAR skill scores to validate the LJ as a tool to forecast heavy rainfall.

A good relationship between heavy rainfall and convective conditions was observed, once again, in the hourly evolution of POD and FAR (Figure 12). The maximum values of POD (good forecast) were observed between 14 and 17 UTC when the temperatures of the day were the highest (diurnal cycle of convection). From this time, the values of POD decreased a little until 0.85, while FAR increased, reaching 0.4. The majority of the thunderstorms, which affected Catalonia during summer months, usually developed in midday and could remain until night due to the high temperatures during all the afternoon (Figure 12). In the first part of the day, a pattern was not found, although the values of POD decreased, reaching the lowest values, around 8 and 9 UTC.

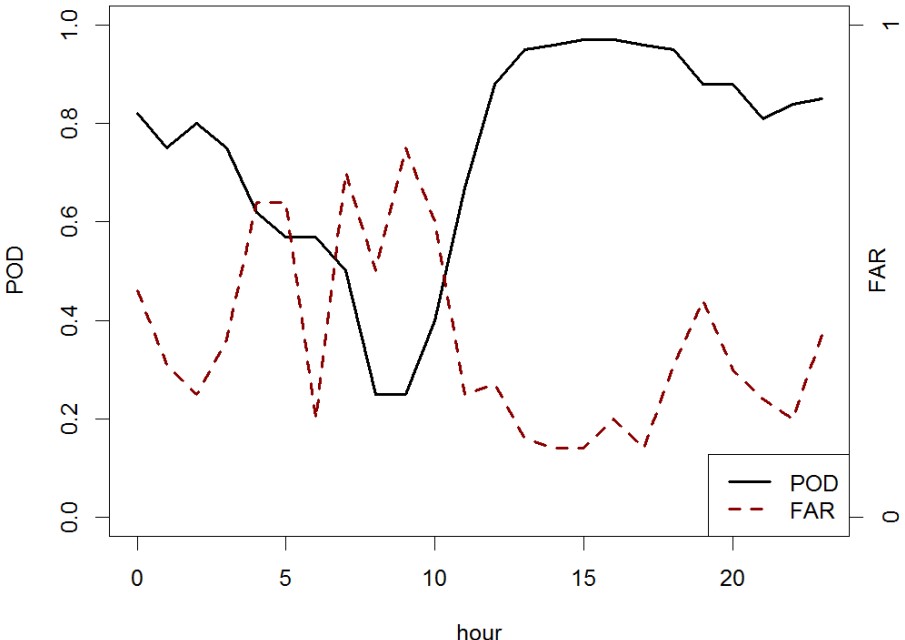

**Figure 12.** Hourly evolution of POD and FAR skill scores to validate the LJ as a tool to forecast heavy rainfall.

Figures 6 and 7 partially explain the results of Figures 11 and 12. The skill scores showed a better performance coinciding with the rise of the number of LJ-producing thunderstorms (June to August, 14 to 19 UTC). Furthermore, it is noted that only a reduced number of cases with LJ warnings did not produce heavy rain records, confirming the deep connection between the two phenomena. An internal analysis of the climatology area of the SMC confirmed this fact. They found that nearly 90% of the events with rain rates over 20 mm/30 min coincided with severe weather phenomena, for the years 2018 and 2019.

The objective of the present analysis was understanding the behavior of the LJ algorithm in front of heavy rainfall, focusing on the particular case of Catalonia. This region suffers many episodes of floods, similar to other areas of the Mediterranean Basin [60–62]. Most of the cases are flash-flood type, which is characterized by heavy rainfall in a short period (less than 3 h) and a high amount of convective rain [35,63]. The previous authors showed that this type of flood was more common in catastrophic events, with a large number of damages and even fatalities. Besides, some previous studies showed that flash-floods were associated exclusively with convective structures, mainly mesoscale convective systems or organized convective-type modes [31,37]. However, the main point is the vertical development of the thunderstorms. In this sense, similar convective modes can produce the same amount of precipitation with a lower electrical activity [37]. On the one hand, those events with shallow convection or warm rain processes [64,65] will produce a large number of false alarms, and they are more frequent in late summer and early autumn. Summer deep convection is the opposite

case. Heavy rainfall clouds also produce high total lightning rates. Then, these events will have the most suitable skill scores.

The end of the analysis was a brief study of the lead time and the distance of those cases with heavy rainfall cells coinciding with lightning jump warnings. The median value of the first parameter was 58 min. This implied that the algorithm was capable of warning one hour ahead. However, it was true that existed a set of 15% of cases in which the warning occurred simultaneously or later than the heavy rain. Regarding the distance, the median value was 34 km (this was the space where the thunderstorm moved between the jump and the rainfall). Again, and corresponding to those simultaneous cases, the distance was less than 3 km in the worst situations.

## 4. Discussion

A classification of the heavy and short duration rainfall events was done according to their relationship with LJ occurrence, obtaining three different groups: $ev_{NOLJ}$ (events without any relationship with LJ), $ev_{PRLJ}$ (events partially related to LJ), and $ev_{LJ}$ (events directly associated with LJ). The two last categories were produced mainly between June and August, while their probability of occurrence decreased notably between September to April. On the contrary, the first group presented major activity during September and October.

Considering the monthly distribution of the IC flashes per event, this presented different shapes depending on the event group. For instance, $ev_{LJ}$ showed a constant and larger median number between April and November. On the contrary, the other two categories presented similar values during all the year, which were notably lower than the events with LJ. Regarding CG flashes, $ev_{LJ}$ contained the higher rate of flashes per event compared to $ev_{NOLJ}$ and $ev_{PRLJ}$, but in this case, the probabilities of occurrence were similar along the year for all classes. This implied that it was possible that the occurrence of a rainfall event exceeded 40 mm/h with low or null activity in all months, but also during the warmest months, the number of cases associated with an LJ were in the majority.

Taking into account the radar data, we only considered the groups $ev_{NOLJ}$ and $ev_{LJ}$. The second group presented mean values of the height of the centroid, which ranged from 2.5 km (cold season) to 8 km (between June and September). Furthermore, TOP12 of the thunderstorms producing $ev_{LJ}$ could reach 15 km, while maximum values of reflectivity were around 60 dBZ. Comparing these values with the thunderstorms producing $ev_{NOLJ}$, the differences were clear, the median centroid height being around 1.5 km (maximum of 5.5 km between July and August, probably associated with warm rain or shallow convection episodes; see [64,66]). Another interesting radar parameter was VIL, which coincided with the other variables in its behavior. This is, $ev_{LJ}$ reached values of 60 mm in summer months, while the maximum value for $ev_{NOLJ}$ was lower, reaching in a few cases 50 mm (August and September). To sum up, all the radar parameters shown in Figure 10 confirmed two facts: firstly, the highest intensity (associated with the reflectivity parameters) and the largest vertical development (TOPs, VIL, and height centroid) of those LJ producing structures, in front of the rest.

The last step of the analysis consisted of the validation of the LJ tool for all heavy rainfall events, using the POD and FAR skill scores. Results showed how POD acquired very good values from April to October (with values over 0.8), being especially high between June and August ( 0.95), coinciding with the warmest temperatures in the region. Besides, during the same periods, FAR values were low. Finally, the hourly evolution of both skill scores presented the best values (large POD and small FAR) coinciding with the hours of the diurnal cycle of convection (14 to 17 UTC). All these results allowed confirming the LJ as a good forecaster of heavy rainfall events in those cases associated with deep convection. On the contrary, shallow clouds producing large rain rates are impossible to forecast, simply because of the flash rate being null or scarce.

## 5. Conclusions

The lightning jump algorithm was revealed as a good forecast tool for severe weather, but also heavy rainfall episodes. However, the research showed that a high dependence on precipitating cloud characteristics existed. In this sense, this resulted in being efficient only in those episodes with well-developed thunderstorms. The reason was that these clouds had high total lightning flash rates associated with them. Then, the best results were directly related to warmer seasons and the period of the day.

**Author Contributions:** Conceptualization, C.F. and T.R.; methodology, C.F. and T.R.; software, C.F.; validation, C.F.; formal analysis, C.F.; investigation, C.F. and T.R.; writing, original draft preparation, C.F.; writing, review and editing, T.R.; visualization, C.F. and T.R. All authors read and agreed to the published version of the manuscript.

**Funding:** This research received no external funding.

**Acknowledgments:** The authors want to thank the colleagues of the Climate, the Meteorological Observation Systems, and Forecast areas, for the data provided and the helpful comments. Besides, we want to thank the four anonymous reviewers for their interesting suggestions.

**Conflicts of Interest:** The authors declare no conflict of interest.

## Abbreviations

The following abbreviations are used in this manuscript:

| | |
|---|---|
| LJ | Lightning Jump |
| POD | Probability Of Detection |
| FAR | False Alarm Ratio |
| IC | Intra-Cloud flash |
| CG | Cloud-to-Ground flash |
| TOP | Radar echo Top |
| VIL | Vertical Integrated Liquid |
| UTC | Coordinated Universal Time |
| LT | Lead Time |
| SMC | Servei Meteorològic de Catalunya (Meteorological Service of Catalonia) |
| XEMA | Xarxa d'Estacions Meteorològiques Automàtiques (Automatic Weather Station Network) |
| AWS | Automatic Weather Station |
| XDDE | Xarxa de Detectors de Descàrregues Elèctriques (Lightning Location System Network) |
| DE | Detection Efficiency |
| LF | Low Frequency |
| VHF | Very High Frequency |
| XRAD | Xarxa de Radar de Catalunya (Catalan Radar Network) |
| QPE | Quantitative Precipitating Estimation |

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
