# Peer review of "The Lightning Jump Algorithm for Nowcasting Convective Rainfall in Catalonia"

_atmosphere, doi:10.3390/atmos11040397_

Round 1

Reviewer 1 Report

The authors present a method of using a lightning based indicator of severe weather to identify heavy rainfall.  This application is a contribution to the scientific field that has not been fully explored, and the reason the reviewer feels it should be accepted pending major revisions.  The reviewer feels that the story the authors are trying to convey through this article is incomplete, and that key pieces of information, if included, would strengthen the work.  The reviewer did not do a complete grammatical review of the article.  The reviewer asks the authors to be quantitative where they can be (numbers versus qualitative descriptions), and present both sides of the algorithm performance.  Below are some aspects that the reviewer felt must be addressed for publication. 

  • 19-20…be specific on what characteristic of the updraft is responsible for the increase in charge separation. Fall speeds of crystals and graupel are far smaller than the speeds of the maximum updraft for lightning jumps. 

  • 30-33 You may find this recent paper in Atmosphere of use in being specific in relating the lightning to precipitation properties as well given the focus of this article is lightning-precipitation relationships

https://www.mdpi.com/2073-4433/10/12/796

  • 37-49 Is the focus on total rainfall, or instantaneous rainfall rate lead time? Its not clear based on the introduction text, although the reviewer is suspecting rainfall rate. Otherwise total rainfall (whether that is 1 hour, 24 hour, etc.) will always lag lightning. The reviewer brings this up because in section 2.3 they mention hourly rainfall totals, but it seems that the jump would then always lead the hourly total.

        Introduction or discussion: There needs to be inclusion on how heavy            rainfall can occur in the absence of strong ice processes. The reviewer          notes that this analysis is geared toward Spain, but there should be              mention on where the lightning will not provide any sort of lead time            given that this analysis can be applied elsewhere around the world.              Please see the CHUVA campaign from Brasil: http://chuvaproject.cptec.inpe.br/portal/noticia.ultimas.logic;jsessionid=0B10F52E453D7C8C99DDD00686C7DB83

A good summary of the lightning-rainfall patterns in Brasil are noted here: https://agupubs.onlinelibrary.wiley.com/doi/10.1029/2019JD030950

      Brasil is a good marker for lightning rainfall relationships because it has        a wide variety of regimes, many of which were studied during the TRMM        validation era in the late 1990s. Does the region in question have heavy        rainfall events in the absence of lightning?  If so, is there a reference or        known frequency at which these events occur that can provide the                reader an idea of how dominant convective rainfall is versus events              driven by warm rain processes across Northeastern Spain? This is                hinted at in the conclusion section on line 366, but needs to be more            explicitly stated in the work.

  • Methodology Bullet Number 2 – Research has long shown that relationships updraft and downdraft process are on the time scale of 30-60 minutes (e.g.,

Dye et al. 1983; https://journals.ametsoc.org/doi/abs/10.1175/1520-0469%281983%29040%3C2083%3ADMEOAC%3E2.0.CO%3B2

Dye et al. 1986; https://agupubs.onlinelibrary.wiley.com/doi/10.1029/JD091iD01p01231,  

Goodman et al. 1988; https://agupubs.onlinelibrary.wiley.com/doi/abs/10.1029/GL015i011p01185)

and the relationships between updraft and lightning jump is approximately on the order of a few minutes (e.g., Schultz et al. 2017; https://journals.ametsoc.org/doi/pdf/10.1175/WAF-D-15-0175.1 ). The question that needs to be addressed in this work is how are processes 150 minutes after a lightning jump physically related to the storm processes which generated the lightning jump? 

  • Lead time is mentioned in the background section, but then is never brought up again throughout the work. How does the lightning jump generate unique information which warns of heavy rain versus solely using lightning flash density?

  • Discussion on the physical mechanisms or weather patterns for the reversal of the POD and FAR between 5 and 10 UTC is necessary. The authors hint at this on lines 355-363, but why is this time period distinctly different from other periods like 14-17 UTC?  (Hint: what is usually different about storms during this morning period versus the afternoon period).

Minor Revisions

Line 99 – How many of the stations produce rainfall totals at fixed estimates vs the 1 minute rainfall totals?

line 100 – minute not minutely.  Also instead of “fixed minutes”, consider “fixed intervals.”

Figure 10 – Given the temporal variation of these parameters, would it be useful to examine the radar parameters by season or time of day?

Line 356 – No need to redefine POD and FAR acronyms

Author Response

Dear reviewer,

thank you very much for your comments. We have tried to include all in the new version. Please, find attached our response. We appologize because it has been impossible to us to copy your suggestions (it seems that your pdf was protected).

Best regards

Reviewer 2 Report

General Comments:

This paper is interesting and is absolutely worthy of publication after some English clean-up. Sections 1,2 and and 5 need to be thoroughly reread by the authors and in places adjustments need to be made for phrasing. I have listed several instances below but I believe I may have missed some instances. The awkward phrasing at times made the first two sections confusing. I believe after some clean up the paper will read much smoother and the heart of the paper will be set up better. 

In general the figures in this paper are great. Specifically, figure 6 and 7 are fantastic! They do a great job communicating the information and are eye-catching. There are many multi- component figures in this paper. The content of these figures are excellent, but the axes need to be adjusted. The axes need to be larger and more readable. Even when zooming very close on the PDF the axes and labels were too small. 

Specific Comments: 

Author text in bold and “quotations”

Reviewer text in plain formatting following a dash (-)

Line 29: Can you please explain how Percent Correct is calculated in either a short formula or a few words?

Line 90: “these element”  - change to these elements

Line 92: “plus other” - change to plus others

Line 95: “or in the way how the rainfall is measured and recorded” -  awkward phrasing. Line Recommend:  or in the way rainfall is measured and recorded 

Line 98-99: “Most of the AWS is capable of recording 1-minute rainfall, which allows generating 30-minute moving window accumulation easily” recommend: Most of the AWS is capable of recording 1-minute rainfall, which easily allows generating a 30-minute moving window accumulation.

Line 101-102: “As it will show later, the AWS data has combined with radar data in order of generating the more realistic hourly rainfall fields possible.” Awkward phrase. Recommend: As it will show later, the AWS data has been combined with radar data in order to generate more realistic hourly rainfall fields.

Line 136: “Weather radar data have used in this work for two purposes.” - recommend: Weather radar data have been used in this work for two purposes. 

Line 150-151: What was the reasoning for selecting 3mm and 25 flashes per day as the bottom threshold for cases? Is this supported by previous studies or was this determined through an iterative process with your dataset? Thank you for the clarification.

Figure 2 (between line 170 and 171): The label on top part of the figure (figure 2a) is being cut off. I recommend reformatting this image so that the labels are fully readable. 

Line 183-84: “it is possible starting the other steps of the analysis. ” - Awkward recommend:  it is possible to start the other steps of the analysis.

Figure 4 (between 189 and 190): Move legends either completely outside or inside the circle.The line going through the legend is distracting. 

Figure 9: The figure is hard to read because of the size. Recommend increasing the size of the figure and increasing the labels/tick marks of the axes.

Figure 10: Increase labels on each axis to improve readability. 

Line 326: “A classification of the heavy and shortly rainfall events has been done” - awkward recommend: A classification of the heavy and short duration rainfall events have been done

Author Response

Dear reviewers,

please, find attached the answers to your suggestions, which have result very useful for the improvement of the manuscript.

We hope you agree with the new version of the document

Best regards

The authors

Reviewer 3 Report

The paper presents an analysis of intense rainfall episodes in Catalonia with and without lightning jumps. It is a very promising hypothesis and an additional reason for monitoring lightning jumps, which could have broad, global implications. It covered a five year time period and therefore was able to build some interesting statistics. Unfortunately, the methodology behind what I view as the crux of the paper due to its largest operational implications-the skill scores-is unclear therefore the results are difficult to evaluate. Please see the specific comments below that can improve the impacts of this study. 

Major comments:

Section 2.2.2: I believe the Information on LLS should be expanded since it is one of the key datasets in the paper. How are locations or CG/IC categorization determined? Are there any established accuracy and detection characteristics of ICs? 

Lines 172 or 117: It is not precisely clear what defines a LJ in this study. Is it from IC or CG flash rates? Is it precisely as in Farnell et al., 2017 or more similar to those in Lines 13-14? Please state directly and reiterate the key methodologies here as they are important in the interpretation of the results and might restrict potential comparisons to other LJ studies. The method in the Farnell et al. study uniquely defined the LJ and noted that their definition did better with severe thunderstorms than simple or multi-cells, so how might the restrictions of the method influence this analysis?

Section 3.3: There is interesting discussion about the seasonal differences, but only the totals are shown in Fig. 10. Would it be possible to display the populations in Fig. 10 more similarly to Fig. 9 in order to support this discussion?

Section 3.4: The discussion of POD and FAR is very promising and is certainly the most immediately applicable result of the study to forecasting (Line 314), so the details are especially important to document in this manuscript. Which dataset were they calculated from? The same as Section 3.3 or any LJs as in the earlier analysis? Is it really useful for forecasting given the large maximum spatial and temporal differences between events (e.g. Line 174)? Does this include LJs occurring after the large rainfall event? The second major issue with this analysis: was the FAR calculated from only the data subset with the high rainfall threshold as suggested in the Section 3 introduction? In other words, are there LJ events without high rainfall rates which are not being included in this analysis? That would have significant implications for the false alarm rate. Similarly if there were rainfall rate events without lightning, would they be included in the POD? These details are very important in determining the robustness of these statistics.

Section 2: It would be very useful to define what is meant by with, without linked and without LJ mean in the methodology section given that Sections 3.1 and 3.3 use different subsets, and be specific and which ones are being discussed where (e.g. Lines 196-200). These are tricky to keep track of when reading the paper. Maybe it would be worthwhile to define a separate abbreviation for the combined ev_pr and ev_nolj subset used in Section 3.3 to help keep it obvious which is being discussed where.

Section 3-4: It would be helpful to the application of the statistical results if some hypotheses were also presented as to why these might exist throughout the paper. For example, are the hourly distributions (Lines 221-227) different because of different storm modes? Are the non-linked LJ associated with larger stratiform areas which might be causing large rainfall rates far from the convective cores and therefore highest lightning rates? These extensions if made throughout the document would help the interpretation of the analyses done. Similarly, it might help to directly frame the analysis as demonstrated when LJs may or may not be helpful in predicting large rainfall events.

Fig. 8: Why not show the same months for each of the lower two sets of plots? According to Fig. 6 they should both have a relatively large number of events in the same months which would make the comparison easier to interpret. This would give strength to the argument that they are produced in different areas as the selected sets might also show climatological differences in storm locations. Also, the text labels are very hard to read.

Minor comments:

Line 40-42: There are many studies which have tried to exploit similar relationships to precipitation rates or cloud volumes in predicting flash rates which seem pertinent to this discussion such as:

Liu et al., 2012 (10.1029/2011JD017123)

Basarab et al., 2015 (10.1002/2015JD023470)

Tippett et al., 2018 (10.1029/2018GL079750)

Romps et al., 2014 (10.1126/science.1259100)

Also please add a citation for non-inductive collisional charging. (e.g. Illingworth and Latham, 1977, Calculations of electric field growth, field structure and charge distributions in thunderstorms, Quart. J. Royal Met. Soc., 103, 281-295 or Takahashi, 1978 10.1175/1520-0469(1978)035<1536:REAACG>2.0.CO;2)

Lines 42-44: That is not directly related to the mixed-phase processes which promote electrification -- the rebounding collisions between ice-phase particles in the presence of supercooled liquid water.

Line 72: Suggest rephrasing to: “Different air masses can affect the study area, mainly from….”

Line 74-75: Suggest rephrasing to: “This, combined with the previous orographic factions, contributes to Catalonia experiencing many severe weather events yearly and also heavy rain episodes…”

Line 77-78: The meaning of this sentence is unclear.

Line 82: “Types of networks”

Line 89: remove “its the main positive point is that” or rephrase.

Line 90: Remove “main” or rephrase “a main” to “the main.” And make element and other (Line 92) plural.

Line 95: “or in how the rainfall”

Line 100: You can use “half-hourly” for the thirty minute interval. Are they switched in the parenthetical examples?

Line 127: Needs citation

Line 155: What is a pixel size?

Line 165: What is the relative frequency of the reflectivity fields used?

Line 174: These seem rather large. Is the temporal range before, after or either?

Line 177-179: It is very hard to understand what is meant.

Line 179: Can an LJ be associated with more than one QPE cell?

Line 190: Which previous step? Is this the single best match from above or a near-event tracking similar to the lightning? Please be specific.

Line 195: What is height centroid?

Line 204: What is meant by “even some warning”?

Line 214: Remove “on the contrary” since it is not conflicting.

Line 233: This would be helpful to have a comment in the caption, as well, that these are representative cases.

Line 235-238: It is also worth noting that lightning, in general, is more common over land than large bodies of water, so LJs might be less common over Mediterranean. (e.g. Albrecht et al., 2016 10.1175/BAMS-D-14-00193.1). Also the plot seems to show a somewhat even distribution over the area whereas the text seems to suggest a maximum over the Mediterranean. Maybe it would stand out more with a different colorbar than used in the other panels?

Line 246: “That the terrain plays”?

Line 251: I suggest rephrasing to “with respect to the intra-cloud (IC) flashes there are two main points. First...”

Line 254: “the second point”

Lines 258-260: I do not understand what is meant by this statement.

Line 263: Is there no lightning activity? Wasn’t having some flashes during the day part of the criteria for selecting flashes?  Of course, they might not be associated with the event itself, but the statement seems to contradict the methodology.

Line 267-270: This is worded confusingly.

Line 275: It is still unclear what the height of the thunderstorm centroid is.

Line 286: What are TOP12 and TOP45?

Line 291: There are better references for the theory of non-inductive collisional charging and connections to flash rates (see above).

Fig. 1: Black rectangle non apparent in Fig. 1 A

Table 1: This table may be unnecessary as only one category is used here.

Fig 2: Multiple (a) labels are present. Contoured features and text labels are hard to see. Suggest making text larger and maybe contours larger or more different from each other? 

Fig. 4: These are total flashes (IC and CG) or just CG?

Fig. 5: There is no reference to Fig. 5 in the text?

Fig. 6: It would be helpful to be consistent between the legend and the caption.

Fig. 10: Do any of these populations contain statistically significant differences?

Author Response

Dear reviewer,

please, find attached the answers to your suggestions, which have result very useful for the improvement of the manuscript.

We hope you agree with the new version of the document

Best regards

The authors

Reviewer 4 Report

See the attached .pdf file

Author Response

(The authors gave the same response as above.)

Round 2

Reviewer 1 Report

The major flaw with the method is the low POD and higher FAR for events in the overnight period. Discussion and inference of the reasons behind this deficiency in the method must be included in the work and in the abstract. Dont just reference other references because this is a two part problem.  What types of convection specific to Catalonia are responsible for the low POD?  What types of convection specific to Catalonia increase the far as high as 0.7?  Its seems like a few more sentences are needed to fully explain the phenomenon because the same storm types/air masses may not be responsible for the different behaviors.  

Lines 80-86, please use common air mass definitions https://sciencing.com/six-types-air-masses-8045253.html

http://glossary.ametsoc.org/wiki/Airmass_classification

105-111 - Again, be specific on the number of stations. "some few" stations is not descriptive. Is it 1%, 10%? More? If its 1% why are these stations being included? 

Really trying to push the justification for using a 1 hour time period given that flash flooding occurs on time scales less than an hour and the event may already be occurring by the time its identified. The main point of the LJ is to generate lead time on a phenomenon, and not just identify occurrence.  Otherwise save a step and just use flash rate. The reviewer is pushing you for a reason...because you have the data availability to do this and take advantage of the lightning jump's strength of early identification.  Seems odd that the higher resolution information is available, but not being utilized and the reason the reviewer keeps hitting home on this point. 

Author Response

Dear reviewer,

thank you very much for your quick response, with very interesting comments. We have tried to give answer to all your new suggestions.

Please, find attached the answers

Best regards 

The authors

Reviewer 3 Report

I thank the authors for the work they put into the revisions and for the comments into consideration.The paper and methodology are much clearer and improved. Most of my following critiques are very minor wording changes and additional clarifications which could further improve the document. However, I am still very puzzled as to how the POD and FAR ratios reported in the study were generated (see explanation below). They are a very significant result, so it is especially important that their generation is methodologically sound.

Major

If there are 14,000 events without a linked lightning jump and 3,800 with one (Intro to Section 3), I cannot understand how the POD can be so large. The POD should be the number of hits (correct forecasts of the LJ warning) divided by the total number of positive events (rainfall events). An estimate for Month 09 based on Figure 6 and the totals in Lines 253-258 would be roughly 0.4x3,800 or ~1500 events with no LJ; 0.1x10,400 or ~1000 without a linked LJ; and only 0.2x3,800 or ~800 events with a LJ, resulting in only 800 hits out of 3,300 events or a POD of a rainfall event by a LJ of roughly 1/4, whereas Figure 11 has a POD close to 1. If it was not the POD that is being shown, or if there is some other way it was calculated than with the numbers above as a reader assumes, this needs to be explicitly described. As it is, the high POD given the values presented previously is highly questionable. 

Minor

Line 19: “rapid intensification”

Line 46: Clarify that some of these studies were looking specifically at LJ and some at total flash rates. Or move the ones looking at total flash rates to the following sentence as other studies which show the link between lightning and rainfall.

Line 48: “rebounding”

Line 50: These are two different types of precipitation processes (cold vs. warm)

Line 75: Replace “accidents” with “features” or the like.

Line 80: Replace “of” with “from”

Line 121-122: I might be worth rephrasing to specifically mention that it does not find altitudes. Not allowing 3D locations could be interpreted differently.

Line 143: Suggest rephrasing: “several points per lightning flash” It seems peculiar that the presumably redundant information would better track the convective cells. Is there a reference for this, yet?

Line 148: Which portions of the study use one point per flash and which use multiple points per flash?

Line 216: Thank you for adding this additional context. I do not understand, however, what is meant by “the terrain event.”

Line 239-241: Should that be “criteria to that used in Point 2”? I also suggest replacing “in this case” with “to radar characteristics” or something similar for clarification.

Line 277: I’m unsure of what “the rest giving out near 49.5%” means.

Line 288-290: I may have miscommunicated my previous point here. Since lightning, in general, is less common over the Mediterranean, it makes sense that a rainfall event is more likely to occur without high lightning rates and therefore a LJ. The authors might additionally choose to expand this discussion by using this figure and section to describe where LJs are less expected or useful for monitoring rainfall events. This is not necessary but would improve the takeaways from this section.

Line 311: I suggest changing “case” to “class” to be consistent with the previous sentence.

Line 320: Is there a reference for this statement?

Line 355: “non-inductive electrification”

Line 394: Reference Fig. 12

Figure 9: Should the bottom row be ev_lj?

Author Response

(The authors gave the same response as above.)
